# Interactive single-cell data analysis using Cellar

Euxhen Hasanaj [1], Jingtao Wang[2], Arjun Sarathi[3], Jun Ding [2✉] & Ziv Bar-Joseph [1,3✉]

Cell type assignment is a major challenge for all types of high throughput single cell data. In many cases such assignment requires the repeated manual use of external and complementary data sources. To improve the ability to uniformly assign cell types across large consortia, platforms and modalities, we developed Cellar, a software tool that provides interactive support to all the different steps involved in the assignment and dataset comparison process. We discuss the different methods implemented by Cellar, how these can be used with different data types, how to combine complementary data types and how to analyze and visualize spatial data. We demonstrate the advantages of Cellar by using it to annotate several HuBMAP datasets from multi-omics single-cell sequencing and spatial proteomics studies. Cellar is open-source and includes several annotated HuBMAP datasets.

[1] Machine Learning Department, School of Computer Science, Carnegie Mellon University, Pittsburgh, PA 15213, USA. [2] Meakins-Christie Laboratories, Department of Medicine, McGill University Health Centre, Montreal, QC H4A 3J1, Canada. [3] Computational Biology Department, School of Computer Science, Carnegie Mellon University, Pittsburgh, PA 15213, USA. ✉email: jun.ding@mcgill.ca; zivbj@andrew.cmu.edu

A number of large consortia including the Human Bio-Molecular Atlas Program (HuBMAP)[1] are focused on profiling tissues, organs, and the entire human body at the single-cell level. These consortiums use several different technologies for studying the molecular composition of single cells including single-cell RNA Sequencing, single-cell ATAC Sequencing[2], single-cell spatial transcriptomics[3], and single-cell spatial proteomics[4]. In addition to these large consortia, individual labs also generate data using some or all of these modalities.

Over the last few years, a number of methods have been developed for the assignment of cell types in single-cell data[5–10]. In most cases, different groups from the same consortia, and even the same group when processing multiple types of single-cell data, rely on a different set of tools. This makes it hard to integrate and compare data from these groups since researchers often use different assignment techniques, markers, and even cell-type naming conventions.

To enable large-scale collaborations, integration, and comparisons across many different single-cell omics platforms and modalities, we developed Cellar, an interactive and graphical cell-type assignment web server. Cellar implements a comprehensive set of methods, both existing and new, which cover all steps involved in the cell-type assignment process. These include methods for dimensionality reduction and representation, clustering, reference-based alignment, identification of differentially expressed genes, intersection with functional and marker sets, tools for managing sessions and exporting results, as well as a dual mode for analyzing and comparing two datasets simultaneously. As cell-type assignment often requires user input in the form of domain knowledge, Cellar adopts a semi-automatic solution that permits users to intervene and modify each processing step as needed. To enable such interactive analysis, Cellar provides methods for semi-supervised clustering and projection of expression clusters in spatial single-cell images. Figure 1 provides an overview of Cellar's workflow. Cellar was tested by members of HuBMAP over the last year and used to annotate several single-cell datasets from different organs, platforms, and modalities.

## Results

**Analysis of scRNA-seq data.** We used Cellar to analyze 11 HuBMAP seq datasets (10x genomics) with an average of 7500 cells from five different tissues (Kidney, Heart, Spleen, Thymus, Lymph node)[11], all of which are available in Cellar. Cellar first performs quality control by removing unreliable cells and low-count genes. Additional normalization and scaling is applied based on user criteria. Cellar then clusters a lower-dimensional representation of the data and further reduces the dimension for visualization purposes. We demonstrate this basic pipeline by analyzing a spleen dataset with 5273 cells (Cellar ID: HBMP3-spleen-CC2). We used PCA, followed by UMAP[12] for dimensionality reduction and the Leiden algorithm[13] for clustering to obtain a total of 16 clusters (Supplementary Fig. 1a). For each cluster, Cellar identified top differential genes. Using the top 500 differential genes, functional enrichment analysis (GO, KEGG[14], MSigDB[15]) identified cluster 0 as B-cells (for example, "B-Cell Activation" (q value = 0) and "B-Cell Receptor Signaling Pathway" (q value = 0) were the top categories for GO and KEGG, respectively). This assignment is further supported by visualizing the concurrent expression of two known B-cell markers *CD79A* and *TNFRSF13C*[16].

In addition to unsupervised clustering, Cellar also implements methods for supervised assignment based on a reference dataset. These can directly utilize the dual mode and other methods implemented in Cellar. For example, this form of assignment can be used in conjunction with Cellar's semi-supervised clustering option to correct noise during the label transfer process. To illustrate such use, we applied Scanpy's Ingest function[17], which is available in Cellar, to integrate two expert-annotated spleen datasets (Cellar IDs: HBMP2-spleen-2 and HBMP3-spleen-CC3). We used HBMP3-CC3 as ground truth and transferred labels from it to HBMP2-2. We then compared the results of label transfer with the ground truth annotations for HBMP2-2 and observed an adjusted rand score (ARI) of 0.39. In contrast, running Leiden clustering on HBMP2-2 leads to a much lower ARI score of 0.27. We then refined the results of label transfer by using a semi-supervised adaptation of Leiden where the least noisy clusters were chosen as constraints and not allowed to change during the iterations of the algorithm. This led to a much better ARI score of 0.66 demonstrating the benefits of label transfer and semi-supervised clustering. These results are shown in Supplementary Fig. 2.

**Analysis of scATAC-seq data.** While scRNA-Seq is currently the most widely used data modality, several other molecular data types are also being profiled at the single-cell level. To illustrate the use of Cellar for such data we used it to annotate scATAC-seq[2]. Cellar can handle scATAC-seq data in two different ways: cell-by-gene and cell-by-cistopic. The former is based on the open chromatin accessibility associated with the nearby region of all genes while the latter relies on cisTopic[10] which uses Latent Dirichlet Allocation[18] to model cis-regulatory topics. The resulting cell-by-gene or cell-by-cistopic matrix is used for downstream analysis such as visualization and clustering. We used Cellar to annotate a scATAC-seq dataset profiling Peripheral Blood Mononuclear Cells[19] (Cellar ID: PBMC 10k Cell-By-Gene) using the cell-by-gene representation. Results are presented in Supplementary Fig. 3. DE analysis for clusters 0 and 4 identified the *KLRD1* marker for natural killer (NK) cells[20].

**Analysis of spatial transcriptomics data (CODEX).** In addition to sequencing assays, recent imaging assays can also provide information on the expression of genes or proteins at the single-cell level. Cellar can be used to analyze such data by providing a side-by-side view of the expression clusters and spatial organization. To illustrate this, we analyzed CO-Detection by indEXing (CODEX)[21] spatial proteomics data. We used a lymph node dataset that contains 46,840 cells (Cellar ID: 19-003 lymph node R2). The clustering results are shown in Fig. 2 along with the corresponding tile for these cells with the projected cluster annotations. Given the small number of proteins profiled in this dataset (19), not all clusters could be assigned to unique types, though several have been assigned based on DE gene analysis in Cellar. Cellar matches the cell colors in the clustering and spatial images, making it easier to identify specific organizational principles and their relationship to the profiled cell types. The spatial tile in Fig. 2 shows that B cells cluster tightly together and are surrounded by T cells and other cell types in the lymph. The B-Cell clusters also contain a subset of proliferating cells.

**Joint analysis of multiple modalities.** Finally, we used Cellar to jointly analyze data from two different modalities. For this, we used a SNARE-seq[22] kidney dataset which profiled both the transcriptome and chromatin accessibility of 31,758 cells (Cellar IDs: kidney SNARE ATAC/RNA 20201005). Here we first ran cisTopic on the chromatin modality and determine cluster assignments by running Leiden on the inferred cis-regulatory topics (Fig. 3a). We use these labels to visualize the expression data in Fig. 3b. This can be easily achieved using Cellar's dual mode, which allows a cell ID-based label transfer from one

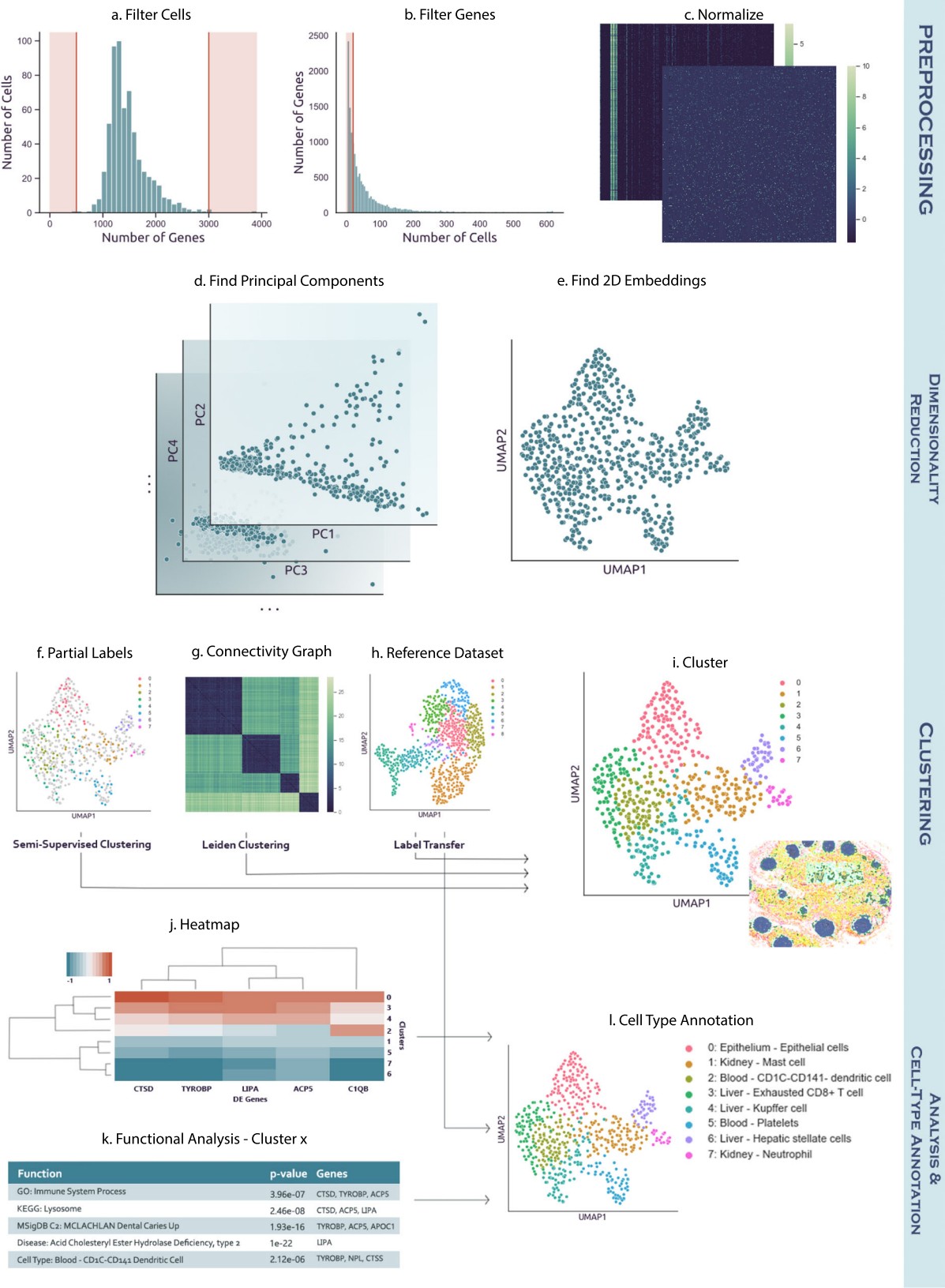

**Fig. 1 Cellar's workflow. a–c** Preprocessing (optional). Cellar can filter cells based on the number of expressed genes, and genes which are rarely expressed. Next the input is normalized. **d, e** Dimensionality reduction and visualization. Several methods for dimensionality reduction are implemented as part of Cellar. The reduced data is then visualized by running another (possibly the same) dimensionality reduction method. **f–i** Clustering. Cellar supports several unsupervised and semi-supervised clustering methods. It also implements supervised label transfer methods. **j–l** Cell-type assignment. Cellar enables the use of several functional annotation databases for the assignment of cell types.

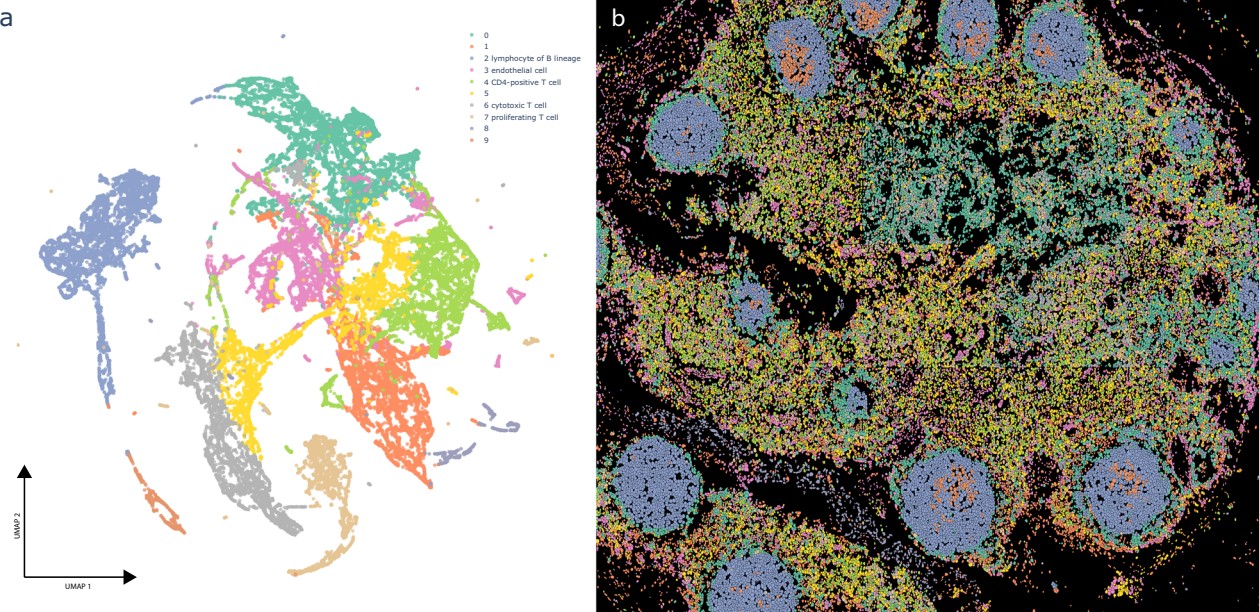

**Fig. 2 CODEX data analysis in Cellar.** (ID: 19-003 lymph node R2) (**a**) UMAP visual representation of a lymph node CODEX dataset with 46,840 cells, clustered via Leiden. **b** Projection of the assignments on the spatial CODEX image that can be visualized side by side in Cellar. Cluster assignments were copied from (**a**). Not all clusters could be assigned to unique cell types given that only a few ten protein levels are measured, though several have been assigned based on differential gene analysis in Cellar. The B-Cell clusters are surrounded by T-cells and other cells types in the lymph. The B cell clusters also contain a subset of proliferating cells.

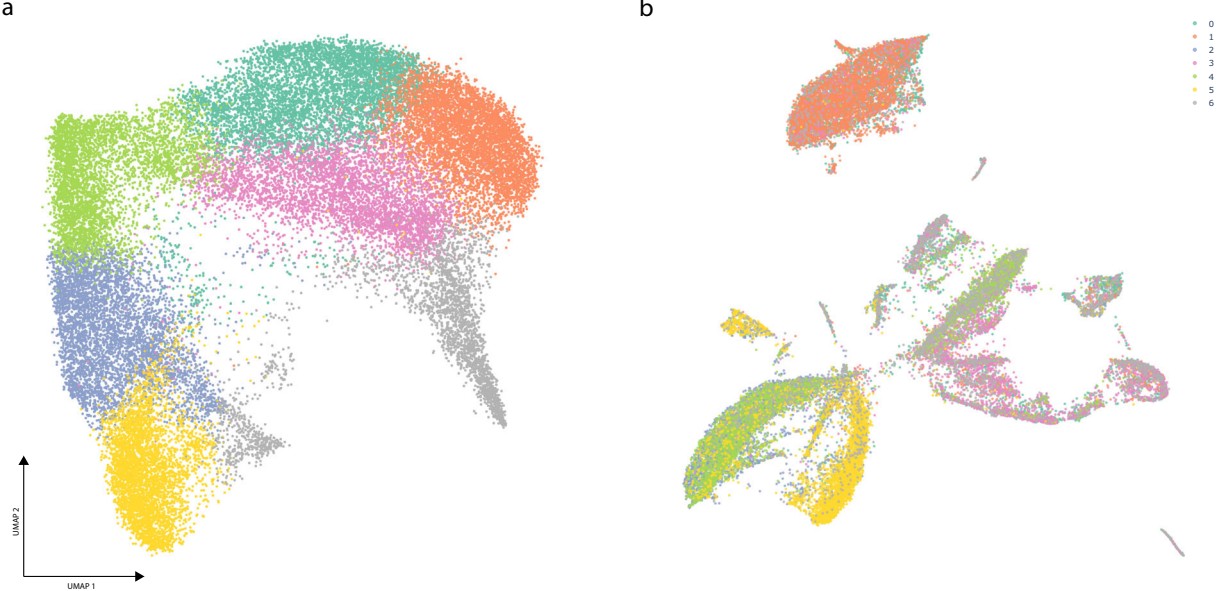

**Fig. 3 SNARE-seq data analysis in Cellar.** (IDs: kidney SNARE ATAC/RNA 20201005) (**a**) UMAP plot of the chromatin modality for the kidney SNARE-seq dataset with 31,758 cells. First, we obtain a cell-by-cistopic matrix by running cisTopic which is then used to define clusters via Leiden clustering. **b** Corresponding UMAP plot of the expression matrix with cluster assignments copied from a. Cellar's dual mode allows a cell ID based label transfer from one modality to the other.

modality to the other. Cellar identified differential genes, and we used these to map cell types. For example, cluster 1 was assigned based on both known markers (*SLC5A12*, p-value = 0) and GO term analysis ("Apical Plasma Membrane", $p$ value = 1e-4), which signify the presence of Proximal Tubule Cells[23,24].

## Discussion
To conclude, Cellar is an easy-to-use, interactive, and comprehensive software tool for the assignment of cell types in single-cell

studies. Cellar is written in Python using the Dash framework and includes efficient operations and data structures for dealing with large datasets. These include using the Annotated Data object[17] in memory-mapping mode which allows the analysis of large datasets by using little system memory, approximate nearest neighbors based on faiss[25] to speed up neighbors graph construction for Leiden clustering, as well as several interactive components for maximum flexibility. Cellar supports several types of molecular sequencing and imaging data and implements several popular

methods for visualization, clustering, and analysis. Cellar has already been used to annotate single-cell data from multiple platforms and tissues. These annotated datasets (mostly from HuBMAP) can serve as a reference for transferring labels to other datasets. For tissues not currently supported by our HuBMAP annotated datasets, Cellar provides several external functional enrichment datasets that, combined with the user's knowledge about specific markers, help in assignment decisions. We hope that Cellar will improve the accuracy and ease of cell-type assignment in single-cell studies. A web server running Cellar can be accessed at https://cellar.cmu.hubmapconsortium.org/app/cellar.

## Methods

Complete details on all the methods used to process, analyze, visualize and integrate the data are available in Supporting Methods.

**Preprocessing.** Preprocessing of the data was done via scanpy[17]. For all scRNA-seq data we filtered cells with less than 50 or more than 3000 expressed genes. We also filtered genes expressed in less than 50 or more than 3000 cells. The data matrix was then CPM total count normalized (total count = 1e5) and log1p-transformed. Finally, we scale the data down to unit variance and zero-mean.

The PBMC scATAC-seq dataset was converted to a gene activity score matrix by summing peaks which intersect the nearby region of all genes as listed in GENCODE v35[26]. The gene ranges were extended with 5000 base pairs downstream and 1000 base pairs upstream. The resulting cell by gene matrix was then normalized and log1p-transformed as explained above.

We did not normalize any of the CODEX data.

**Clustering, visualization, and functional analysis.** scRNA-seq and gene activity matrices were reduced to a 40 dimensional space via PCA. We used the PCA implementation of the scikit-learn package with a randomized SVD solver. The lymph node CODEX data was reduced via UMAP[12] with 10 dimensions using the Python package umap-learn. The embeddings were then used to construct an approximate neighbors graph using faiss[25] with 15 neighbors, and then clustered using the Leiden community detection algorithm for graphs[13] with a default resolution of 1. Only for the lymph node CODEX data we used a smaller resolution of 0.1 in order to obtain a reasonable number of clusters. All data was reduced from these embeddings to 2 dimensions using UMAP for visualization purposes.

Differential gene expression analysis was performed with diffxpy (https://github.com/theislab/diffxpy) by using a Welch's t-test. The 500 DE genes with the greatest fold-change values were selected for enrichment analysis via the package gseapy (https://github.com/zqfang/GSEApy) which uses the GSEA method[27]. Only for the CODEX data, where the number of channels was small (<20), we used all differentially expressed proteins found.

**Label transfer and semi-supervised clustering.** Label transfer between HBMP2-spleen-2 and HBMP3-spleen-CC3 was performed using scanpy's Ingest (https://scanpy.readthedocs.io/en/stable/generated/scanpy.tl.ingest.html). Ingest projects the query dataset to a latent space fit on reference data using PCA with 40 components. We only consider overlapping genes between the two datasets. Following label transfer, we use semi-supervised Leiden (resolution = 1) to refine the cluster assignments, where clusters 0, 4, 9, 10 were "frozen" (see Supplementary Fig. 2c for a scatter plot of the aforementioned clusters). The ARI score was computed on ground truth annotations assigned by a human expert. For the unconstrained version of Leiden used in the experiment we also set a default resolution of 1.

**Joint analysis and cisTopic.** The SNARE-seq data was formed by combining four separate kidney SNARE-seq datasets. We removed cells for which no annotations were found. The chromatin modality was processed using cisTopic[10] to discover 40 topics. This number was selected via cisTopic's log-likelihood model selection method. These topics were then treated as a reduced version of the data and used for clustering and visualization in the same way as described earlier for scRNA-seq data.

**Reporting summary.** Further information on research design is available in the Nature Research Reporting Summary linked to this article.

## Data availability

All data analyzed in this study are available for download from the application's web server as well as the HuBMAP portal at https://portal.hubmapconsortium.org with access codes:

HBMP468.VQQQ.574 [https://portal.hubmapconsortium.org/browse/dataset/14946a8eb12f2d787302f818b72fdc4e], HBM536.GZQR.922 [https://portal.hubmapconsortium.org/browse/dataset/35a639b983ff85728bdb3cbe0eac360a],

HBM279.SLFX.335 [https://portal.hubmapconsortium.org/browse/dataset/3f678ab5cd7ed086ec0d2d4468fc5094], HBM695.NCKX.893 [https://portal.hubmapconsortium.org/browse/dataset/800f1703d81373c58ca5ca0b76e52d79], HBM684.ZPCL.638 [https://portal.hubmapconsortium.org/browse/dataset/91a46cd9228c04e77df05536a036824b], HBM595.QDQD.996 [https://portal.hubmapconsortium.org/browse/dataset/6ab211e1ca46633ceaeba0ae6f385538], HBM327.JDHF.334 [https://portal.hubmapconsortium.org/browse/dataset/bafcb8882be3c213101755a0468f3620], HBM476.NNFJ.275 [https://portal.hubmapconsortium.org/browse/dataset/4a682d67bea887e5bb1ade2bd137489e], HBM638.GFJG.839 [https://portal.hubmapconsortium.org/browse/dataset/9ad464b54a0087db94bfca405d4ad968], HBM894.XCHW.375 [https://portal.hubmapconsortium.org/browse/dataset/851d76f833dc8dc3debb8eb2d73d543b], HBM437.KPNV.984 [https://portal.hubmapconsortium.org/browse/dataset/6d094503393d6d41b6193c9d10e33d9e, HBM439.NZNH.823 [https://portal.hubmapconsortium.org/browse/dataset/65b92f0191dc73e9470f46ceb217054d]. We also use public data from the 10xGenomics website with the id atac_v1_pbmc_10k [https://www.10xgenomics.com/resources/datasets/10-k-peripheral-blood-mononuclear-cells-pbm-cs-from-a-healthy-donor-1-standard-1-1-0]. See Supplementary Table 2 for a full list of access codes and IDs.

## Code availability

Code is available from the GitHub repository: https://github.com/euxhenh/cellar/[28]. Full documentation is available at https://euxhenh.github.io/cellar/.

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

## Acknowledgements

This work was partially supported by NIH grants OT2OD026682, 1U54AG075931, and 1U24CA268108 to Z.B.J. J.D. was supported by Fonds de recherche du QuÂbecâ SantÂ (FRQS) -Junior 1. The results here are in whole or part based upon data generated by the NIH Human BioMolecular Atlas Program (HuBMAP).

## Author contributions

Z.B.J., E.H., and J.D. designed the software. E.H. developed and implemented the back-end including dimensionality reduction, clustering, cell-type annotation, and data integration methods. E.H., J.W., and A.S. contributed to the implementation of the front-end interactive visualizations and also contributed to the implementation of enrichment analysis of identified signature genes. All authors contributed with manuscript writing. All authors read and approved the final manuscript.

## Competing interests

The authors declare no competing interests.
