## [Peer Review File · Nature Communications]

Interactive single-cell data analysis using CellarReviewers' Comments:

Reviewer #1:

Remarks to the Author:

In this manuscript, the authors developed a webserver for cell-type assignment of single-cell data, named Cellar. Cellar provides functions covering all the steps in cell-type annotation, including preprocessing, dimensionality reduction, clustering, marker gene identification, and label transfer-based cell type assignment. In addition, the workflow includes functions for cell-type assignment in single cell ATAC-seq data and spatial proteomics data. Cellar also has a "side-by-side" visualization mode for integrated analysis of data from two different data modalities.

Cell-type assignment is critical for single-cell data analysis and many tools have been developed for it, as have been cited in ref 5-10 in the manuscript. The workflow in Cellar is mainly based on existing algorithms, with an addition of the "interactive" part, in which a semi-supervised clustering is implemented for users to define the clusters they would like to keep before another round of clustering. The authors claims that the semi-supervised step would improve the quality of clustering and thus cell-type annotation, but few data support was provided.

1. In order to show that the semi-supervised clustering steps improved the quality of cell-type assignment, the authors shall directly compare the accuracy of cell-type annotation with the existing annotation tools using a set of benchmark data.
2. It is not addressed how the supervised annotation of data, which still heavily rely on domain knowledge of users, will promote sharing and comparison of data from different groups of users. Is it possible for Cellar to provide a more defined protocol/guideline for the users to annotate data with a set of well-benchmarked methods and name the cell types with a controlled vocabulary?
3. In Figure S2, it is not clear how the silhouette score was calculated. From definition of silhouette score, a score of -0.03 to 0.1 represents very poor separation between clusters. It is not known if such an increase in number represented significant improvement of clustering.
4. The "side-by-side" mode in Cellar is useful for viewing the multimodality data, but the workflow did not actually include integrative analysis tools for multimodality data. Such tools are available in softwares such as Seurat 3 [1], which used cells with shared molecular features as "anchor" points for different modality of datasets. Including these tools in the Cellar server will definitely increase the utility of the webserver.
5. For the writing of the paper, the Supporting methods, Supporting results and Supplementary Figures may be moved into the main text.

Reviewer #2:

Remarks to the Author:

This manuscript describes Cellar, an interactive tool to perform many types of single-cell annotation analyses. Cellar implements popular methods for each step in the single-cell annotation analysis pipeline. It is able to analyze scRNA-seq, scATAC-seq, and even imaging data. The tool contains datasets from the HuBMAP consortium that can be used for annotation and exploration.

I played with the tool and it seems to be doing the job. I think that the UI/UX can be improved, as I had a few times I wasn't sure what to do and namings weren't always clear. Anyhow, I can see why members of HuBMAP find it useful for their research.

The only difficulty I have with this manuscript is that there is nothing novel here. It's a nice tool and

the description of tools used can also help more advanced users in annotation analyses to decide how to perform an analysis. As a tool, there are many interactive single-cell analysis tools (see list under "Interactive visualization and analysis" in <https://github.com/seandavi/awesome-single-cell>). I think it might be a good idea to turn this manuscript into a review article for annotation tools and suggest to users "best practice" pipeline for research (that can be facilitated with Cellar).

Minor comments

1. "In addition to these large consortia, several individual labs generate data using some or all of these modalities" – weird to use the term "several" here, there are hundreds of individual labs. Only a small fraction of the single-cell data is coming from non-consortia studies.

2. I didn't see any word on the implementation. What language Cellar is written, what resources are required for running it, computational limitations (I know the answers, just expected to see it in the text).

Reviewer #3:

Remarks to the Author:

In this manuscript, Hasanaj et.al described a new tool Cellar (Shiny app) for interactive annotation of single-cell data. Single-cell annotation is an unsolved difficult problem, many tools have been developed to automatically annotate cell types/states, but they are far from being perfect. Expert opinions are needed from biologists to accurately annotate the cell clusters. Being able to annotate the cell clusters in a semi-supervised way is critical. What's nice about this tool is that It can also be used for imaging data and scATACseq data.

There are some issues that need to be addressed:

1. What's the scalability? The max number of cells used in the example is ~35,000, how does it scale up for millions of cells. I am worried about the interactive performance and memory usage.
2. The authors implemented label transferring with SingleR. Seurat CCA performs well, better to include it.
3. The purpose of being able to annotate the clusters in a semi-supervised manner is to use the expert opinion for better cluster annotation. In Line33, it says "merging 0,3,7 as B cells". They could be different B cells. Also, in line 58, it says "merging 0,3,4 clusters" They could be different NK cells (based on CD56, CD16, and CD57 expression gradient)! Simply merging them actually is defeating the purpose.
4. For the web app, there needs to be more documentation for each functionality. I saw some sparse documentation at https://github.com/ferrocactus/cellar/blob/master/doc/cellar_guide.md
5. One of the demanding features users want is to be able to compare multiple samples across different groups (treated vs control). It will be nice to have those features. The users will be able to upload a metadata sheet to describe the comparisons they want to make.
6. The authors may need to mention what are the advantages and disadvantages with the cellxgene (<https://github.com/chanzuckerberg/cellxgene>) and commercial solutions such as 10x browser and the bioturing browser <https://bioturing.com/>
7. Line 38, missing a comma after "clustering". Line 39, missing a comma after "this".

Reviewer #4:

Remarks to the Author:

Review:

This paper presents Cellar, a graphical user interface web server for interactive cell type annotation in various types of high-dimensional single cell data.

This work doesn't present any algorithmic innovations, rather its main advantage is the claimed convenience of wrapping several existing tools in a web-based UI

That's where the main problem is. I was not able to figure out how to use Cellar in order to reproduce the results in the paper. The UI seems less than intuitive to use and there are occasional bugs that prevented me from moving forward. I strongly feel that the Cellar application needs a) tutorials that guide the user step-by-step to every figure in the paper b) better testing. Additionally, I highly recommend hiring a skilled frontend developer to help flushing out the UI. R/Shiny has a lot of bugs and performance limitations and gets very tricky when you move beyond simple dashboards and try building a sophisticated application with a lot of interacting and inter-dependent parts.

Among the bugs that I observed directly:

Uploading the sample CODEX data and pressing 'regenerate tile'. I pressed the button 10 times by accident and the tile kept re-generating 10 times over the next few minutes

Loading CODEX HMBP_lymph_node_1.h5ad data -> going to Pre-processing-> pressing 'run' with default settings throws an error 'data has negative values'

Running dim. reduction with UMAP shows a UMAP plot with a standard Plotly library. While the plot allows for selecting cells using rectangular/lasso selection, I can't see any way to turn the selected cells into a population and look up its expression profiles, etc. Also I can't see a way to color this plot by expression values, which would be

Not sure how this plot will scale with 100K or a 1M cells

Running Leiden clustering colors the plot by clusters. However, I cannot find a way to select individual clusters and look at their gene expression levels. DE tab is empty. Violin plot tab contains a slider, but nothing else.

Heatmap tab works, but selecting genes seems to put a lot of strain on the CPU, each selection event takes about a second and renders my (relatively beefy) computer unresponsive. It seems unusable Moving on to Constrained Clustering, selecting 'Constrained K-means' with default settings and pressing 'Run' doesn't do anything unless dimensionality reduction has been run first, but there is no warning/notification

Running label transfer using SingleR and using CODEX_Florida_lymphnode_19_003 dataset hung up the application completely. The screen greyed out and it needed to be restarted, losing all the analysis The application doesn't have any state management - it seems that when you reload the page, you lose all the analysis and need to start from scratch

Additionally, the authors claim that the server is intended to be used for large-scale collaborations and data integration, but no dataset used in the paper is particularly large. It would be worthy demonstrating that Cellar can handle some of the larger datasets out there (> 1M cells) Also, how many users can the server handle concurrently? Does it auto-scale the number of cloud instances depending on the workload?

I also have specific comments regarding the CODEX data:

Unclear where the data came from (certainly not from Goltsev et al. 2018 which is the only CODEX paper cited in the text) and what panel was used, was it fresh-frozen or FFPE tissue etc.

It's unclear how does the algorithm used to predict cell types in CODEX using classifiers trained scRNA-seq data account for differences in scaling and expression patterns between the two very

different types of data, and also how it matches protein markers to gene names, which don't always map the same way (for instance, CD45RA, CD45RO and CD45 all map to the same gene PTPRC, but different isoforms thereof)

With the data provided in the paper, I am unable to evaluate the performance of the cell type annotation, as there is no comparison given to any kind of gold standard cell type annotation. Also couldn't figure out how to do that comparison in Cellar UI

In general, the cell type predictions provided by Cellar in CODEX data seem rather coarse and general, it could only predict the identity of less than half of the clusters, and also those identities are rather basic (B-cells, T-cells etc), which raises the question about the real-world utility of Cellar for CODEX and other spatial types of data. I feel like Cellar needs to incorporate a CODEX-specific training set instead of attempting to transfer the labels from models trained on scRNA-seq data

We thank the reviewers for their comments. Below we provide a detailed, point by point, response to the comments.

Reviewer #1 (Expertise: scRNASeq annotation):

In this manuscript, the authors developed a web server for cell-type assignment of single-cell data, named *Cellar*. *Cellar* provides functions covering all the steps in cell-type annotation, including preprocessing, dimensionality reduction, clustering, marker gene identification, and label transfer-based cell type assignment. In addition, the workflow includes functions for cell-type assignment in single cell ATAC-seq data and spatial proteomics data. *Cellar* also has a “side-by-side” visualization mode for integrated analysis of data from two different data modalities.

Cell-type assignment is critical for single-cell data analysis and many tools have been developed for it, as have been cited in ref 5-10 in the manuscript. The workflow in Cellar is mainly based on existing algorithms, with an addition of the “interactive” part, in which a semi-supervised clustering is implemented for users to define the clusters they would like to keep before another round of clustering. The authors claims that the semi-supervised step would improve the quality of clustering and thus cell-type annotation, but few data support was provided.

- 3. In order to show that the semi-supervised clustering steps improved the quality of cell-type assignment, the authors shall directly compare the accuracy of cell-type annotation with the existing annotation tools using a set of benchmark data.*

As suggested we have added analysis to demonstrate that semi-supervised analysis improves the accuracy of the assignments. For this, we used semi-Supervised Leiden (SSL) which is an extension of the Leiden algorithm that incorporates constraints of the form “cell A belongs to the same cluster as cell B.” The membership for these “must-link” points is set in advance and is not allowed to change during the iterations of SSL. We used several annotated datasets to test the usefulness of the SS method implemented by *Cellar*. We first ran a grid search on the resolution parameter and determined the value for which vanilla Leiden achieves the highest ARI score. We then run SSL with the same resolution value and select at random x% of the points to include as must-link constraints. Finally, we compute the ARI score for both Leiden and SSL (to allow for a fair comparison, we removed the points that were used as constraints). Figures below present the ARI scores when the % of constrained cells varies from 0% to 50% of the total number of cells. We see that with as little as 10% of the cells given as constraints, the ARI score improves by up to 15% on several of the Spleen datasets we analyzed (Figure S4).

2. It is not addressed how the supervised annotation of data, which still heavily rely on domain knowledge of users, will promote sharing and comparison of data from different groups of users. Is it possible for Cellar to provide a more defined protocol/guideline for the users to annotate data with a set of well-benchmarked methods and name the cell types with a controlled vocabulary?

We agree with this comment, which was also a major reason for the development of Cellar. Cellar will address the challenge of common annotations using ontologies curated by HuBMAP. HuBMAP has an ongoing effort to establish an agreed upon set of ontologies for cell types in the tissue it is profiling (<https://hubmapconsortium.github.io/ccf-asct-reporter/>). Cellar integrates these ontologies and restricts assignments for all modalities (scRNA-Seq, scATAC-Seq, spatial single cell) to the approved set of types guaranteeing better agreement between data from different groups and labs. In addition, as we discuss Cellar supports annotations using cell type specific markers. These will also be directly obtained from the HuBMAP cell type marker list for the different cell types (available at the same URL as above). While these are not mandatory, they will help users agree on which cells should be assigned to which types.

3. In Figure S2, it is not clear how the silhouette score was calculated. From definition of silhouette score, a score of -0.03 to 0.1 represents very poor separation between clusters. It is not known if such an increase in number represented significant improvement of clustering.

We initially computed the silhouette score on the PCA embeddings and euclidean distances between vectors in this space tend to be similar, hence, the low silhouette score. To avoid confusion, we have replaced the silhouette scores with more appropriate metrics. In particular, we switched to the ARI score for the datasets for which we have ground truth annotations.

4. The “side-by-side” mode in Cellar is useful for viewing the multimodality data, but the workflow did not actually include integrative analysis tools for multimodality data. Such tools are available in softwares such as Seurat 3 [1], which used cells with shared molecular features as

“anchor” points for different modality of datasets. Including these tools in the Cellar server will definitely increase the utility of the webserver.

As the reviewer suggests, in addition to unsupervised clustering Cellar also implements methods for supervised assignment based on a reference dataset. These can directly utilize the dual mode and other methods implemented in Cellar. For example, this form of assignment can be used in conjunction with Cellar's semi-supervised clustering option to correct noise during the label transfer process. To illustrate such use, we applied the Scanpy's Ingest function which is available in Cellar, to integrate two expert-annotated spleen datasets, HBMP2 2 and HBMP3 CC3. We used the HBMP3 CC3 dataset as ground truth and transferred labels from it to the HBMP2 2 dataset. We then compared the results of label transfer with the ground truth annotations for HBMP2 2 and observed an adjusted rand score (ARI) of 0.39. We next selected the least noisy clusters for HBMP2 2 as constraints for the Semi-Supervised Leiden algorithm and used these to obtain a new clustering for the dataset. This led to a much better ARI score of 0.66. In contrast, running vanilla Leiden with no constraints on HBMP2 2 leads to a much lower ARI score of 0.27 demonstrating the benefits of label transfer and semi-supervised clustering. These results were added as Figure S2.

5. For the writing of the paper, the Supporting methods, Supporting results and Supplementary Figures may be moved into the main text.

We thank the reviewer for this comment. We submitted this as a 'brief communication' and so were severely constrained in terms of the text and figures we could include. We will consult with the editors and if they agree will be happy to move more info from the supplement to the main text.

Reviewer #2 (Expertise: scRNASeq user/analysis/annotation):

This manuscript describes Cellar, an interactive tool to perform many types of single-cell annotation analyses. Cellar implements popular methods for each step in the single-cell annotation analysis pipeline. It is able to analyze scRNA-seq, scATAC-seq, and even imaging data. The tool contains datasets from the HuBMAP consortium that can be used for annotation and exploration.

I played with the tool and it seems to be doing the job. I think that the UI/UX can be improved, as I had a few times I wasn't sure what to do and namings weren't always clear. Anyhow, I can see why members of HuBMAP find it useful for their research.

We agree that the UI, which was based on the R package Shiny, was not optimal for this tool. Based on this comment we completely redesigned and re-implemented Cellar. In this new version of Cellar, we switched to Plotly's Dash framework for building the interface and the UI components. Dash offers many advantages, such as improved performance, better error handling, as well as a multi-threaded environment that allows the execution of multiple Cellar components at the same time (provided they are independent). This is ideal when using Cellar in dual mode as users can now run clustering/analysis for both their datasets simultaneously. The new version of Cellar is written in Python, with an interface for R libraries. As the reviewer can see, it is much faster and more efficient than the original version. The new interface was also designed to be more intuitive to use as well as requiring fewer clicks to achieve a desired goal as compared to the previous version of Cellar.

The only difficulty I have with this manuscript is that there is nothing novel here. It's a nice tool and the description of tools used can also help more advanced users in annotation analyses to decide how to perform an analysis. As a tool, there are many interactive single-cell analysis tools (see list under "Interactive visualization and analysis" in <https://github.com/seandavi/awesome-single-cell>). I think it might be a good idea to turn this manuscript into a review article for annotation tools and suggest to users "best practice" pipeline for research (that can be facilitated with cellar).

We thank the reviewer for this comment, and based on it we have improved the description of the novel features as well as added more examples of their usefulness. Specifically, we view the following as unique to Cellar

1. The ability to perform interactive analysis (for example, the constrained clustering option which as we now discuss improves the accuracy of cell type assignment). We have added to the Supplement new results for running constrained clustering on a single dataset and running constrained clustering following label transfer, in both cases showing significant improvement in accuracy. See Figures S2 (also below) and S4.

A: Ground truth annotations for the HBMP2-2 spleen dataset. B: Cluster assignments after running vanilla Leiden with default parameters (ARI: 0.27). C: Transferred labels from HBMP3-CC3 using Scanpy Ingest (ARI: 0.39). D: Cluster assignments after running Semi-Supervised Leiden with default parameters on top of transferred labels from C (ARI: 0.66). The constrained clusters were 0, 4, 9, 10.

2. The fact that the Cellar tool is providing common ontology and marker selection for annotating data from several different modalities including scRNA-Seq, scATAC-Seq, Spatial Transcriptomics and Spatial Proteomics. We are not aware of any other tool that provides support for the annotation of single cell data from all these modalities.
3. Unique features to support assignment of single cell spatial transcriptomics and proteomics data. Our new design provides an interactive visualization and analysis platform that combines both the clustering in the expression space (via the standard tools for analysis of single cell expression data) and a projection of the clusters into spatial space to visualize the spatial organization of the annotated cell types.
4. In the new version we have also implemented several methods to speed up the processing of large single cell datasets including the use of a multi-threaded environment. We have also included in the revised version approximate nearest neighbor algorithms which significantly improve clustering (Leiden) run time. These

algorithms, implemented in package faiss (<https://github.com/facebookresearch/faiss>) are used for the leiden clustering, semi supervised leiden, spectral clustering and assignment confidence calculations. The approximated version has almost identical results as the original one but is much faster. (reducing the time for average size datasets from 5 minutes to 10 seconds).

5. Finally, the linkage of the tool to HuBMAP will provide a unique opportunity for both HuBMAP and non HuBMAP researchers to perform joint analysis of single cell data. All HuBMAP data will be available through Cellar and users would be able to upload their own data to directly compare, align or annotate using Cellar and the processed HuBMAP data.

Minor comments

1. "In addition to these large consortia, several individual labs generate data using some or all of these modalities" – weird to use the term "several" here, there are hundreds of individual labs. Only a small fraction of the single-cell data is coming from non-consortia studies.

We are not sure if the reviewer meant 'Only a small fraction of the single-cell data is coming from consortia studies' instead of what is written, since most data is still generated by individual labs. In any case we removed the word 'several' from the revised version.

2. I didn't see any word on the implementation. What language Cellar is written, what resources are required for running it, computational limitations (I know the answers, just expected to see it in the text).

We thank the reviewer for this comment and added a section to the Supplement with implementation details. Briefly, Cellar is written in Python, with few components using R libraries. With the new memory-mapping mode, little memory is needed to run Cellar despite the size of the datasets. However, for better performance we recommend at least 8GB of RAM and a multi core CPU (to speed up algorithms such as PCA, UMAP, and clustering which greatly benefit from parallelization). We provide a dockerized version of Cellar, thus running Cellar is system independent assuming that docker is installed on that system.

Reviewer #3 (Expertise: sc-data analysis, including scATAC-Seq):

In this manuscript, Hasanaj et.al described a new tool Cellar (Shiny app) for interactive annotation of single-cell data. Single-cell annotation is an unsolved difficult problem, many tools have been developed to automatically annotate cell types/states, but they are far from being perfect. Expert opinions are needed from biologists to accurately annotate the cell clusters. Being able to annotate the cell clusters in a semi-supervised way is critical. What's nice about this tool is that It can also be used for imaging data and scATACseq data.

There are some issues that need to be addressed:

1. What's the scalability? The max number of cells used in the example is ~35,000, how does it scale up for millions of cells. I am worried about the interactive performance and memory usage.

This is a good point and we now provide examples in which Cellar analyzes datasets with hundreds of thousands of cells for both scRNA-Seq and codex (spatial proteomics) data. We have re-designed Cellar and it is now able to quickly process and store such datasets. First, we now use memory-mapping to load the datasets, which is a mechanism that maps the file on disk to a range of addresses within the app, thus making analysis possible with little RAM usage. This allows us to support a much larger number of users as well as larger datasets. Second, we also included a faster variant of Leiden clustering which uses approximate neighbors to construct the connectivity graph, thus greatly reducing processing time with little loss in accuracy.

To test this we used data from the idiopathic pulmonary fibrosis atlas (IPF_atlas on Cellar) from Adams et al. (<https://advances.sciencemag.org/content/6/28/eaba1983>) which contains about 350,000 cells. After preprocessing, we obtain 240,837 cells which we reduce and then cluster. The run times are shown below (using an Intel Core i5-6300HQ, 4x 3.2GHz and 16 GB of RAM):

Dimensionality Reduction (PCA with 40 components + UMAP): 4m 42s

Clustering (approximate neighbors + Leiden with resolution=1): 4m

Semi-Supervised Clustering (approximate neighbors + Semi-Supervised Leiden, 15% of the cells given as constraints): 30s

We further benchmarked the performance of approximate neighbors by computing the overlap between the neighbors obtained via KNN and those obtained via AKNN for several datasets. Table 1 in the supplement and below describes these results as well as the running time for each.

Dataset	# Cells	# Features	kNN (s)	ANN (s)	Overlap (Top 50)
HBMP3 spleen CC2	5273	9549	0.65	0.13	0.96
W101 heart	7717	16651	1.39	0.24	0.93
HBMP2 thymus 2	13203	9645	4.59	0.59	0.90
HBMP1 lymph node 1	14348	9995	5.61	0.67	0.88
CODEX 19-003 lymph node R2	46840	19	2.75	1.09	0.98
IPF atlas	240837	4528	-	23.94	-

2. The authors implemented label transferring with SingleR. Seurat CCA performs well, better to include it.

We could not integrate Seurat’s CCA into the Cellar pipeline due to the different data format that Seurat uses. However, we added another label transformation method, Scanpy’s Ingest, in addition to SingleR.

3. The purpose of being able to annotate the clusters in a semi-supervised manner is to use the expert opinion for better cluster annotation. In Line33, it says “merging 0,3,7 as B cells”. They could be different B cells. Also, in line 58, it says “merging 0,3,4 clusters” They could be different NK cells (based on CD56, CD16, and CD57 expression gradient)! Simply merging them actually is defeating the purpose.

We agree with this comment and in the new version completely replaced the example of constrained clustering to more quantitatively evaluate it. For this, we used semi-Supervised Leiden (SSL) which is an extension of the Leiden algorithm that incorporates constraints of the form “cell A belongs to the same cluster as cell B.” The membership for these “must-link” points is set in advance and is not allowed to change during the iterations of SSL. We used several annotated datasets to test the usefulness of the SS method implemented by Cellar.. We first ran a grid search on the resolution parameter and determined the value for which vanilla Leiden achieves the highest ARI score. We then run SSL with the same resolution value and select at random x% of the points to include as must-link constraints. Finally, we compute the ARI score for both Leiden and SSL (to allow for a fair comparison, we removed the points that were used as constraints). Figure S4 presents the ARI scores when the % of constrained cells varies from 0% to 50% of the total number of cells. We see that with as little as 10% of the cells given as constraints, the ARI score improves by up to 15% on several of the Spleen datasets we analyzed. (Figure S4)

4. For the web app, there needs to be more documentation for each functionality. I saw some sparse documentation at https://github.com/ferrocactus/cellar/blob/master/doc/cellar_guide.md

We thank the reviewer for this comment. Following this and comments from Reviewers 2 and 4 we have completely redesigned the Cellar App making it more intuitive and faster. In addition, we now provide detailed documentation on a Github page (https://jingtaowang22.github.io/cellar_docs/). The Github documentation is extensive and provides both information on the methods and features and screenshots of the analysis, so we are hoping it would make using the tool much more convenient. Finally, we now also provide several tutorials on YouTube for the different types of data that users of Cellar are interested in: <https://www.youtube.com/watch?v=J61itSMezFI&list=PL5sLSLkTYpWgfBQ0M8ObfBlqDMAzx0-D2&index=3> (basic analysis pipeline) <https://www.youtube.com/watch?v=QBUXhFZrHec&list=PL5sLSLkTYpWgfBQ0M8ObfBlqDMAzx0-D2&index=2> (joint analysis) <https://www.youtube.com/watch?v=zG3j3DdqLUQ&list=PL5sLSLkTYpWgfBQ0M8ObfBlqDMAzx0-D2&index=1> (spatial data)

Links to the tutorials, documentation, and the source code can also be found at the bottom of the UI.

5. One of the demanding features users want is to be able to compare multiple samples across different groups (treated vs control). It will be nice to have those features. The users will be able to upload a metadata sheet to describe the comparisons they want to make.

Following this comment we revised Cellar to allow direct comparisons of two datasets in a single viewer. Users can upload two different conditions and analyze them side by side using several different options including clustering, the expression of key genes, functional enrichment etc. In addition, Cellar implements methods for alignment of datasets allowing users to use annotated

references for automatically annotating new data. Switching from one dataset to the other is done with the click of a single button. See also Figure

6. The authors may need to mention what are the advantages and disadvantages with the cellxgene (<https://github.com/chanzuckerberg/cellxgene>) and commercial solutions such as 10x browser and the bioturing browser <https://bioturing.com/>

Response: We agree and added text to both Discussion and the Supplement text that explains the unique features of Cellar w.r.t to the methods listed by the reviewer.

Cellxgene was developed for single-cell transcriptomics data and so does not support the single cell spatial data that Cellar supports. Even for scRNA-Seq there are significant differences. While Cellxgene can be used to visualize, cluster, and annotate cells in such studies, it does not provide direct access to various functional annotation datasets and databases and comprehensive cell type specific marker lists which is very important for aiding expert annotations. Finally, Cellxgene does not support the side by side analysis that is critical for both comparing datasets and integrating them. All of these issues are addressed by Cellar.

As for commercial solutions such as 10x browser and bioturing browser, these are usually tailored to a specific platform and do not work well with data from other platforms or technologies. For example, while Bioturning provides comprehensive support for single-cell transcriptomics data, similar to Cellxgene it does not provide direct access to functional enrichment analysis (e.g., GO, pathway) is not incorporated in BioTurning to help with the cell annotation datasets. Bioturing also does not allow for joint analysis of datasets from multiple studies and multiple types of data types. Therefore, it cannot be used to combine spatial and single cell sequencing data or for label transfer. Furthermore, several of the new methods implemented in Cellar including constrained clustering, approximate nearest neighbor support and more are missing from this tool.

Another, more conceptual, difference between Cellar and previous methods is the interactive options that Cellar implements. Cellar allows users to directly modify the cell clustering/annotation based on their expertise if they are not satisfied with the automatic data analysis. The users are also allowed to impose constraints on clustering/annotation (via Constrained Clustering modules that we implemented). This high “interactivity” between the method and the users is unique for Cellar. For each step of the single-cell data analysis pipeline (visualization, clustering, and annotation), we have implemented several commonly used methods in Cellar. This is in contrast to the fixed pipeline used by most other methods. Thus, Cellar provides much higher flexibility for advanced users.

7. Line 38, missing a comma after “clustering”. Line 39, missing a comma after “this”.

Corrected.

Reviewer #4 (Expertise: scRNASeq/single cell data annotation/single-cell imaging)

Review:

This paper presents Cellar, a graphical user interface web server for interactive cell type annotation in various types of high-dimensional single cell data.

This work doesn't present any algorithmic innovations, rather its main advantage is the claimed convenience of wrapping several existing tools in a web-based UI

That's where the main problem is. I was not able to figure out how to use Cellar in order to reproduce the results in the paper. The UI seems less than intuitive to use and there are occasional bugs that prevented me from moving forward. I strongly feel that the Cellar application needs a) tutorials that guide the user step-by-step to every figure in the paper b) better testing. Additionally, I highly recommend hiring a skilled frontend developer to help flushing out the UI. R/Shiny has a lot of bugs and performance limitations and gets very tricky when you move beyond simple dashboards and try building a sophisticated application with a lot of interacting and inter-dependent parts.

We agree that the UI, which was based on the R package Shiny, was not optimal for this tool. Based on this comment we completely redesigned and re-implemented Cellar. In this new version of Cellar, we switched to Plotly's Dash framework for building the interface and the UI components. Dash offers many advantages, such as improved performance, better error handling, as well as a multi-threaded environment that allows the execution of multiple Cellar components at the same time (provided they are independent). This is ideal when using Cellar in dual mode as users can now run clustering/analysis for both their datasets simultaneously. The new version of Cellar is written almost entirely in Python, with few components using R libraries. As the reviewer can see, it is much faster and more efficient than the original version. The new interface was also designed to be more intuitive to use as well as requiring fewer clicks to achieve a desired goal as compared to the previous version of Cellar.

Among the bugs that I observed directly:

Uploading the sample CODEX data and pressing 'regenerate tile'. I pressed the button 10 times by accident and the tile kept re-generating 10 times over the next few minutes

Since Cellar was fully redesigned this issue no longer exists. Clicking the same button again, will launch a new process that will simply replace the previous one.

Loading CODEX HMBP_lymph_node_1.h5ad data -> going to Pre-processing-> pressing 'run' with default settings throws an error 'data has negative values'

We believe that this was caused because the reviewer was trying to preprocess already processed data, hence, the error. To avoid such confusion in the future, we now check if raw

counts are present in the data file, so that when users click preprocess for a second time, we use the raw data instead. However, if raw counts were not found, we proceed with the main data. Since it is difficult to check if data is preprocessed, we rely on negative values as a signal that the data was scaled (and possibly preprocessed). This means that the users need to be careful about the type of data they upload (whether it has already been preprocessed/normalized or not).

Running dim. reduction with UMAP shows a UMAP plot with a standard Plotly library. While the plot allows for selecting cells using rectangular/lasso selection, I can't see any way to turn the selected cells into a population and look up its expression profiles, etc. Also I can't see a way to color this plot by expression values, which would be
Not sure how this plot will scale with 100K or a 1M cells

When viewing the plot, the lasso selection tool is active by default. You can use the lasso tool to select a group of cells, and then under the "Tools" tab, you can set a name for that group of cells ("Subset Name"). This named subset will then appear on the cluster list under the "DE Analysis" tab which can be used to find DE genes for that specific group of cells only. As for the plot with a large number of cells, we now use the webgl format for the plots, which allows for fast rendering and interaction. As for scalability, we now provide examples in which Cellar analyzes datasets with hundreds of thousands of cells. Specifically, we tested Cellar on a scRNA-Seq dataset with 240,837 cells. The run time for several of the analysis methods we implemented for this data are detailed below (using an Intel Core i5-6300HQ, 4x 3.2GHz and 16 GB of RAM):

Dimensionality Reduction (PCA with 40 components + UMAP): 4m 42s

Clustering (approximate neighbors + Leiden with resolution=1): 4m 1s

Running Leiden clustering colors the plot by clusters. However, I cannot find a way to select individual clusters and look at their gene expression levels. DE tab is empty. Violin plot tab contains a slider, but nothing else.

The redesigned software now provides an easy way to perform this. After clustering is complete, a list of all clusters will be populated under the "DE Analysis" tab. Selecting the desired cluster and clicking "Find DE Genes", should return a table of DE genes for that cluster. Once these genes have been found, they can be selected from the "Feature Visualization" tab where users can view their expression levels, heatmaps, or even violin plots. When multiple genes are selected at once, we apply a "min" operation to their expression values after some appropriate scaling has been performed (see supplement).

Heatmap tab works, but selecting genes seems to put a lot of strain on the CPU, each selection event takes about a second and renders my (relatively beefy) computer unresponsive. It seems unusable

We agreed. We ask the reviewer to try the new implementation which solves this problem and is much much faster.

Moving on to Constrained Clustering, selecting 'Constrained K-means' with default settings and pressing 'Run' doesn't do anything unless dimensionality reduction has been run first, but there is no warning/notification

Clustering this data (20K dimensions) requires dimensionality reduction to be run first. Notifications have been added to notify the user about this.

Running label transfer using SingleR and using CODEX_Florida_lymphnode_19_003 dataset hung up the application completely. The screen greyed out and it needed to be restarted, losing all the analysis

This has been fixed. We used SingleR to transfer labels from CODEX_Florida_20-008-lymphnode#10_righthalf to CODEX_Florida_20-008-lymphnode#10_lefthalf. The results are shown in the figure below. SingleR took about 1 hour to run on these datasets, but we found that Scanpy's Ingest function, that we added to Cellar, is faster while also giving comparably good results.

When an error occurs, the screen no longer grays out and the app maintains its current state, thus preventing any loss of analysis.

A)

A) Reference dataset CODEX_Florida_20-005-thymus-CC1-A_tophalf

B)

B) Transferred labels to CODEX_Florida_20-005-thymus-CC1-A_bottomhalf

The application doesn't have any state management - it seems that when you reload the page, you lose all the analysis and need to start from scratch

We use the h5ad format as session files. The session file can be downloaded by clicking "Export Session" at the bottom of the sidebar. Uploading the session file back to Cellar should recover the clustering / analysis done on the data.

Additionally, the authors claim that the server is intended to be used for large-scale collaborations and data integration, but no dataset used in the paper is particularly large. It would be worthy demonstrating that Cellar can handle some of the larger datasets out there (> 1M cells) Also, how many users can the server handle concurrently? Does it auto-scale the number of cloud instances depending on the workload?

We do not have a dataset with more than 1M cells in HuBMAP at the moment. However, to address the comment about scalability we test a new scRNA-Seq lung dataset with more than 200K cells as discussed above]. As the figure below shows, Cellar is able to handle this dataset and works well for all analysis tasks.

The Reviewer can try the dataset by loading scRNA-seq / IPF_atlas from Cellar. We have also tested codex datasets with hundreds of thousands of cells and they cluster quickly using the new implementation of Leiden which uses the fast approximate neighbors algorithm to construct the connectivity graph which is then used as a starting point for the Leiden algorithm. Again we ask the reviewer to try the following files: CODEX_Florida_19-001-thymus-CC1-A, CODEX_Florida_19-003-spleen-CC3-E.

As for managing the workload, we now use memory-mapping to load the datasets, which is a mechanism that maps the file on disk to a range of addresses within the app, thus making analysis possible with little RAM usage. This allows us to support a much larger number of users.

I also have specific comments regarding the CODEX data:

Unclear where the data came from (certainly not from Goltsev et al. 2018 which is the only CODEX paper cited in the text) and what panel was used, was it fresh-frozen or FFPE tissue etc.

We used the following datasets that are publicly available from the HuBMAP portal (<https://portal.hubmapconsortium.org/>).

Cellar ID : HuBMAP ID

Florida_19_001_thymus_CC1_A : HBM376.QCCJ.269

Florida_19_003_lymph_node_R2 : HBM279.TQRS.775

Florida_19_003_spleen_CC3_E : HBM337.FSXL.564

We also added a table in the supplement with HuBMAP reference IDs for all datasets we used in the paper as well as links to the HuBMAP portal for each dataset.

All information about tissue handling is available on the HuBMAP portal.

It's unclear how does the algorithm used to predict cell types in CODEX using classifiers trained scRNA-seq data account for differences in scaling and expression patterns between the two very different types of data, and also how it matches protein markers to gene names, which don't always map the same way (for instance, CD45RA, CD45RO and CD45 all map to the same gene PTPRC, but different isoforms thereof)

We are not sure if the reviewer means the alignment or cell type transformation algorithms when the reviewer mentions 'classification'. In the case of the data presented in the paper we have not used such alignment methods for codex since, as the reviewer notes, there are critical differences in the types of data (sequencing vs. imaging) between the two. Instead, the codex data in Cellar was annotated by a human expert following clustering by Cellar and then using Cellar to visualize marker genes and their expression in different clusters. In addition, the expert projected the clusters on the spatial representation which further refined the cell type assignment. Thus, the expert only used the unsupervised options for the codex assignment without relying on the alignment, or supervised options. The issue of alignment between scRNA-Seq and imaging data is an open problem which is beyond the scope of this paper..

With the data provided in the paper, I am unable to evaluate the performance of the cell type annotation, as there is no comparison given to any kind of gold standard cell type annotation.

As mentioned, this is a new HuBMAP dataset that was annotated using Cellar by the team that generated the data. There is no ground truth in this case and the main reason for presenting this was to show the usefulness of the different interactive features for expert users when annotating their new data. We did test the constrained leiden clustering for scRNA-Seq data with ground truth annotations and have added results for this analysis below. See Also Figure S4.

Also couldn't figure out how to do that comparison in Cellar UI

We now include a tutorial and a YouTube video that explains how to perform alignment in Cellar.

In general, the cell type predictions provided by Cellar in CODEX data seem rather coarse and general, it could only predict the identity of less than half of the clusters, and also those identities are rather basic (B-cells, T-cells etc), which raises the question about the real-world

utility of Cellar for CODEX and other spatial types of data. I feel like Cellar needs to incorporate a CODEX-specific training set instead of attempting to transfer the labels from models trained on scRNA-seq data.

We agree. However, this is not because of a limitation of Cellar but rather because of limitations of the data. The codex data used to generate the figure included only 18 channels (or, in other words, profiled only 18 proteins). These were not enough to identify all cell types in the sample and also not enough to obtain a fine resolution of cell subtypes for the immune cells profiled. However, when codex datasets with more proteins are profiled (new HuBMAP datasets include between 30 and 50 proteins) we expect the resolution to be much better and the coverage of cell types to increase as well. As noted, Cellar can generalize to more proteins and the ability to use its interactive features to perform the analysis would enable much better characterization of cell types for these new datasets.

Reviewers' Comments:

Reviewer #1:

Remarks to the Author:

The authors have satisfactorily addressed my previous questions in their response letter.

I have no further comments.

Reviewer #2:

Remarks to the Author:

The authors have made substantial work to revamp the tool and the manuscript. The tool now looks (to my non-professional eye) much more user-friendly, and much clearer to work with.

I still had issues when playing with it. Some things didn't work for me - for example, I wasn't able to change to UMAP, and other things, like clustering, took very long. I think that additional QA is needed before going live.

Another suggestion is to do more pre-processing offline, at least for default parameters. Also, it might be a good idea to develop some kind of estimation of time for each procedure and present it to the user.

The dual-mode is very nice.

Reviewer #3:

None

Reviewer #4:

Remarks to the Author:

Overall, the new reworked app is a big improvement in terms of functionality. However, since it has been re-written from scratch, there are still some glitches that need to be addressed become this app may be useful to the general audience.

To begin with, I loaded CODEX_Florida_20-008-lymphnode#10_lefthalf dataset

- 1) Unable to change the colors of populations -> impossible to distinguish cell types in the tile plot, since the default colors
- 2) Need an ability to plot cells by XY with plotly and hover over to find out their cluster assignments
- 3) Plot coloring is very slow and tedious from the UI standpoint. Switching between markers on the plot requires 4 clicks: delete the old marker, add the new marker, plotting -> plot expression -> wait several seconds.
- 4) If multiple markers are selected for plot coloring, unclear what is actually displayed. It just says 'normalized val.'
- 5) DE analysis is very slow for CODEX data, and doesn't work for all clusters (e.g. Florida Dataset LN3, Cluster 1 doesn't show any DE markers, even when Alpha is set to 1.0 (how can that be?). A much better solution would be a line plot of either user-selected set of markers, or top N most variable markers.
- 6) When comparing two clusters (or groups) in DE, it would be nice to see what the median expression of each marker is in both groups, having a column 'Median' is not very useful if there's nothing to compare the values to.
- 7) Unclear how label transfer works. Unlike the old version of CellR, I don't see an option to upload/select a reference dataset. The tab "Label transfer" only offers to select which way to obtain

labels (singlr, ingest, cellID), but not the dataset. When I press "Run", nothing happens.

8) Since label transfer such a central function to the application, it would be nice to have a link directly in the UI, to some reference or tutorial explaining how to use this function.

9) Cluster 9 and in Florida Dataset LN3 are most likely proliferating and non-proliferating B-cells, respectively

10) DE results don't get cleared when datasets are switched

11) Github tutorial is still showing the old Shiny-based UI. This needs to be updated.

12) In the rebuttal, the authors are mentioning a video tutorial. I couldn't find any reference to it in the GitHub documentation

Reviewer #2

I still had issues when playing with it. Some things didn't work for me - for example, I wasn't able to change to UMAP, and other things, like clustering, took very long. I think that additional QA is needed before going live.

We agree with the reviewer that for some datasets clustering can indeed take several seconds. However, this is a limitation of the single cell data where hundreds of thousands of cells are being clustered. One of the major improvements in the revised version was a new implementation of the clustering algorithm using approximate nearest neighbors (rather than exact NN). As we show in Supplementary Table 1, this greatly reduced the runtime of clustering. We believe the clustering performance of Cellar is now one of the fastest, certainly compared to other implementations of Leiden (which is the most popular method for clustering single cells).

We note that other packages that use Leiden can be much slower than Cellar. For example, Scanpy constructs KNNs via UMAP / Exact KNN / or a GPU implementation (<https://scanpy.readthedocs.io/en/stable/generated/scanpy.pp.neighbors.html>) all of which can be slow or require additional hardware (GPU). Approximate neighbors perform faster than these as we have shown. Another popular package, Seurat, also uses approximate neighbors and their method of choice, called "annoy" (<https://satijalab.org/seurat/reference/findneighbors>) is similar in terms of run speed to our method according to this benchmark <http://ann-benchmarks.com/>.

Run time of other clustering methods, including KMeans and Spectral Clustering implemented in Cellar depends on the number of restarts. The more restarts the better result but also slower clustering. We have originally used a default of 10 for the number of restarts but changed it based on this suggestion to 1 to speed up the runtime (users could adjust the number using the setting bar)

Another suggestion is to do more pre-processing offline, at least for default parameters.

We note that Cellar is mainly meant to be a tool for the analysis of new, user uploaded data. While we did provide some datasets from HuBMAP this was mainly for demonstration purposes. Since Cellar expects user data, there is no way to perform offline analysis (as the data is not there). We purposely did not perform such analysis for the datasets we uploaded since we wanted to give the reviewers as sense of how it would be to use Cellar with real user data. The only thing we did do for uploaded datasets is run dimensionality reduction so that loading one dataset would immediately populate the plot and the users can start analyzing it.

Also, it might be a good idea to develop some kind of estimation of time for each procedure and present it to the user.

We agree that this will be a nice additional feature. We provide some estimates in the new version. These can be seen by expanding the "bulb" button at the top left corner of the app.

Method	Est. Runtime
PCA (40 PC) + UMAP	9s
PCA (40 PC) + t-SNE	21s
PCA (40 PC) + PCA	1s
Leiden Clustering (auto + max iter)	4s
DE Analysis (1 vs rest)	14s
Scanpy Ingest (Label Transfer)	18s - 37s
SingleR (Label Transfer)	2m 3s - 4m 5s

Reviewer #4 (Remarks to the Author):

Overall, the new reworked app is a big improvement in terms of functionality. However, since it has been re-written from scratch, there are still some glitches that need to be addressed become this app may be useful to the general audience.

To begin with, I loaded CODEX_Florida_20-008-lymphnode#10_lefthalf dataset

1) Unable to change the colors of populations -> impossible to distinguish cell types in the tile plot, since the default colors

We note that unique colors are already assigned by Cellar to each cluster. Given the number of clusters some may seem similar and so the suggestion by the reviewer would be useful for very large datasets.

First, we have added an option that allows the user to manually set the color palette. The palette can be accessed at the plot's toolbar. The color of any cluster can be changed by simply entering the hex code of the new color and applying the palette. Note: any CODEX tiles may need to be re-generated in order to use the new palette.

Secondly, specific clusters can already be highlighted by double-clicking on the legend at the desired cluster ID. Also, clicking once on any cluster ID will hide that cluster. This should make it easier to filter similar colored clusters and focus on the desired ones (applies to plot only).

2) Need an ability to plot cells by XY with plotly and hover over to find out their cluster assignments

We agree that this will be a nice addition and have now implemented this. We decided to show only the cluster IDs and not the annotations in the CODEX tile since these tiles are typically very high-resolution and adding a string containing the annotation to each pixel tremendously increased the rendering time (not to mention data transfer over the network). The mapping between these cluster IDs and the annotations can be seen at the annotations' menu.

3) Plot coloring is very slow and tedious from the UI standpoint. Switching between markers on the plot requires 4 clicks: delete the old marker, add the new marker, plotting -> plot expression -> wait several seconds.

We thank the reviewer for this suggestion. This is a useful user feedback. We have added functionality to make this easier. Clicking on any DE gene in the DE table will automatically copy that gene into the feature visualization dropdown menu, so that the user no longer needs to type the name of the gene manually. The second functionality we added is a button "DE" next to the feature visualization menu that copies all the DE genes in the visible page to the select menu. This should make it easier to run functions like Heatmap without requiring the user to type each gene separately.

4) If multiple markers are selected for plot coloring, unclear what is actually displayed. It just says 'normalized val.'

This is explained in detail in the documentation https://jingtaowang22.github.io/cellar_docs/docs/ui-components/analysis/feature-visualization.html and the supplement. We also changed the label from "normalized val." to "Min Co-Expression" to clarify what is being displayed.

5) DE analysis is very slow for CODEX data, and doesn't work for all clusters (e.g. Florida Dataset LN3, Cluster 1 doesn't show any DE markers, even when Alpha is set to 1.0 (how can that be?). A much better solution would be a line plot of either user-selected set of markers, or top N most variable markers.

First, we note that Codex is profiling very few proteins (less than 20) and so not all clusters have DE genes. Independent of the value of alpha, we did not consider genes to be DE if their fold-change value

was less than 1, which was the case for Cluster 1. However, we do realize that it is better to allow the user to tune this threshold. We added this option to the DE panel. Setting the FC threshold to 0 will not apply any filtering.

6) When comparing two clusters (or groups) in DE, it would be nice to see what the median expression of each marker is in both groups, having a column 'Median' is not very useful if there's nothing to compare the values to.

We thank the reviewer for this suggestion. We agree that this is useful and have now added columns for the mean of DE gene for both the selected cluster and the rest. These means are computed on the logged values (if any log transform was applied), but the fold-change uses the count values if "Is Data Logged" is set to True.

7) Unclear how label transfer works. Unlike the old version of CellR, I don't see an option to upload/select a reference dataset. The tab "Label transfer" only offers to select which way to obtain labels (singlr, ingest, cellID), but not the dataset. When I press "Run", nothing happens.

As we noted in the submission, we have a video tutorial (linked from the bottom of Cellar, <https://www.youtube.com/watch?v=QBUXhFZrHec>) that shows all steps in label transfer. We also provide detailed documentation in https://jingtaowang22.github.io/cellar_docs/docs/tutorials/tutorial3.html. For this specific question, we only allow Label Transfer under "Dual Mode" in Cellar. i.e., users can upload two datasets under both plots and perform label transfer there. The inactive dataset is the one that serves as reference data.

8) Since label transfer such a central function to the application, it would be nice to have a link directly in the UI, to some reference or tutorial explaining how to use this function.

As mentioned above, we have links in both the UI (bottom part) and the git to detailed documentation about the function. We ask that the reviewer give these a try and see if they help address the questions the reviewer had.

9) Cluster 9 and in Florida Dataset LN3 are most likely proliferating and non-proliferating B-cells, respectively

We thank the reviewer for this comment. These assignments were performed by the group that profiled the Lymph datasets. As noted, Cellar is meant to be used for cell type assignments in new datasets and so the group uploading the data is the one making the assignments. Once a dataset is uploaded to Cellar and is public, other groups can save their own copy and change the assignment as they see fit. However, for now we cannot change assignments for data profiled by HuBMAP. We will discuss these with the Florida group and change the assignments if they agree.

10) DE results don't get cleared when datasets are switched

We thank the reviewer for this suggestion. We now clear the list when a dataset is switched.

11) Github tutorial is still showing the old Shiny-based UI. This needs to be updated.

We have fully rewritten the github documentation and the link on the cover page of the resubmission linked to the new git. We ask that the reviewer use that link and check that the documentation is satisfactory. Here is the link for reference: https://jingtaowang22.github.io/cellar_docs/

12) In the rebuttal, the authors are mentioning a video tutorial. I couldn't find any reference to it in the GitHub documentation

As noted above, the link to the videos is at the bottom of the Cellar interface (click the 'Demo' link). These links are also available in the documentation under the corresponding tutorial section. For reference, here is the link to the playlist:

<https://www.youtube.com/playlist?list=PL5sLSLkTYpWgfBQOM8ObfBlqDMAzx0-D2>

** In a separate email the reviewer suggested that we include two new integration methods: GLUER and STvEA. We thank the reviewer for the suggestion as these methods fit very well into Cellar's framework.

- 1) STvEA was used to map CODEX to a matching CITE-seq dataset. While we do not have CITE-Seq data from HuBMAP at this stage we do expect to get some in the near future so this is very helpful. To address this comment we added support for CITE-seq. This involves being able to load both the gene and protein expression as a single file. We describe the details about the file format here: https://jingtaowang22.github.io/cellar_docs/docs/ui-components/sidebar/tools.html

Second, we added the STvEA integration algorithm. We have a tutorial that describes how to perform this integration in Cellar:

https://jingtaowang22.github.io/cellar_docs/docs/tutorials/tutorial4.html

The integration must be performed under "Dual Mode" as described in the tutorial. After the integration is complete, users should be able to see expression levels of scRNA-Seq for genes mapped from CITE-seq in the CODEX plot. These mapped genes can be found under the "Other" features tab. E.g.,

CD4

We now include the same datasets that STvEA authors used and have uploaded it to Cellar so the functionality can be tested on real data. Running the integration on the full CODEX dataset took about 1 hour, so we provide a smaller ~3,000 cell dataset that can be found under the name "BALBc1_Goltsev_CODEX_3k". The reviewer may wish to test this one instead.

Furthermore, we decided to incorporate STvEAs "adjacency score" that is used to compute co-localization scores between clusters and proteins. This score uses a neighbors' graph that is computed on the spatial locations of the cells and not the protein expression. These scores can be found under the "Spatial Data" section. We also describe them in the documentation https://jingtaowang22.github.io/cellar_docs/docs/ui-components/spatial/coloc.html and the tutorial https://jingtaowang22.github.io/cellar_docs/docs/tutorials/tutorial2.html

- 2) We have not implemented GLUER since it would likely be too slow even with GPU support that is required. GLUER was used to integrate single-cell transcriptomics data with imaging-based spatial proteomics data. The authors claim it took them ~5 minutes for 10,000 cells on a GPU. Since our datasets can have much more than 100,000 cells and since it is currently unlikely that we would be able to provide multiple GPUs to support Cellar, we decided not to provide this for now.